# GradientMix: A Simple yet Effective Regularization for Large Batch Training

## Abstract

Stochastic gradient descent (SGD) is the core tool for training deep neural networks. As modern deep learning tasks become more complex and state-of-the-art architectures grow as well, network training with SGD takes a huge amount of time; for example, training ResNet on the ImageNet dataset or BERT pre-training can take days to dozens of days. To reduce the network training time, distributed learning using a large batch size for SGD has been one of the main active research areas in recent years, but this approach entails a significant degradation in generalization. To address this issue, in this paper, we propose a simple yet effective regularization technique, *GradientMix*, for large-scale distributed learning. *GradientMix* can enhance the generalization in large batch regimes by giving appropriate noise through a mixup of local gradients computed at multiple devices, which is contrary to the conventions that simply average local gradients. Furthermore, *GradientMix* is optimizer-agnostic, hence can be applied to any popular optimization algorithm as long as the overall loss is expressed as the sum of the subgroup losses. Our extensive experiments show the effectiveness in both small and large-scale problems, and especially we consistently achieve state-of-the-art performance for various optimizers on training ResNet-50 on the ImageNet dataset with *32K batch size*.

## 1 Introduction

Stochastic gradient descent (SGD) is a classical training approach commonly used for training most deep learning models, in which model parameters are updated in the opposite direction of the gradient of the loss function computed on a mini-batch. The recent developments in hardware such as GPU and TPU enables several complex and state-of-the-art models to be trained using large batch whose gradient is computed efficiently via data parallelism. Unfortunately, however, it is known that large batch training typically suffers from severe degradation in generalization compared to small batch training. The reason for poor generalization of large batch training still has not been fully uncovered, which leads many researchers to investigate this phenomenon actively in several directions.

For that reason, many efforts have been made to shed light on whether generalization has to do with sharp minima in the perspective of loss landscape. As one of pioneering work, Keskar et al. (2016) argued that the generalization gap is primarily due to the sharp local minima obtained from large batch training. Another study He et al. (2019a) further showed that local minima as well as sharp minima lie on an asymmetric valley and that a local minimum biased towards the flat side generalizes better than the exact empirical minimizer. Despite some evidence of the relationship between sharp minima and generalization, Dinh et al. (2017) substantiated that the performance degradation in large batch training has nothing to do with sharp minima by showing that all minima of the neural networks can be arbitrarily sharp according to the reparametrization that does not change the network output.

Given that there are limitations to enhance generalization using the curvature information of the loss function, designing an optimization algorithm specific to large batch regime arises as another dimension. The first representative in this line of work is LARS optimizer You et al. (2017), which exploits the ratio of the parameter size to gradient size. Yet, as pointed out in Zhang et al. (2020), LARS performs poorly for attention-based model such as Transformer on the ground that LARS is an algorithm based on vanilla SGD. In order to address the issue, the second representative optimizer LAMB is proposed in You et al. (2019), which inherits the spirit of LARS and one of the adaptive gradient methods, Adam Kingma & Ba (2015). The aforementioned optimizers make it possible

to train ResNet-50 on the ImageNet dataset and BERT around an hour by scaling the batch size to more than 32K. However, Nado et al. (2021) empirically confirmed that traditional optimizers such as Nesterov can achieve as high performance as large batch optimizers (e.g. LARS, LAMB, AdaScale SGD, and etc.) if one puts the same effort into the hyperparameter tuning for each optimizer. Owing to such an observation, it is debatable whether the superiority of large-batch optimizers like LARS and LAMB is due to their well-designed concepts or just merely due to dense hyperparameter tuning.

Last but not least, there have been several studies to elucidate how the noise can enhance generalization. Bottou (1991); Neelakantan et al. (2015) showed that noise can play a significant role in training neural networks although they do not focus on large batch training. In addition, Zhou et al. (2019) theoretically demonstrated that adding noise to the gradient is helpful in circumventing a spurious local optimum, thus allowing for converging to a global optimum, and Liu et al. (2021) showed that noisy gradient descent finds a flat minimum in a non-convex matrix factorization. Smith et al. (2020) empirically corroborated that noise in stochastic gradients can enhance generalization and Wen et al. (2018) dipped into how injecting curvature noise can be useful in large batch training.

In the light of previous studies regarding gradient noise, one can say "certain noise can regularize the gradient descent procedure well". However, most of the aforementioned studies on gradient noise are restricted to the Gaussian-type noise. Therefore, it is still questionable whether other types of noise could be beneficial in the aspect of generalization especially for a large batch regime.

In this paper, we focus on the approach to improve generalization for large batch training with the use of the mixup Zhang et al. (2018) technique that has been recently highlighted in the data augmentation context. Toward this, we propose the novel way of adding noise for the optimizer by *mixing* up the gradients and show that this technique in fact encourages convergence to a flat minimum.

Our main contributions with some details are as follows:

- We propose a simple yet effective regularization *GradientMix* for large batch training. *GradientMix* performs a linear combination of local gradients computed at each device (or each sample) with arbitrary mixing rates. Our *GradientMix* is optimizer-agnostic, hence it can be applied to any optimization algorithms including LARS or LAMB for large batch training.
- We mathematically investigate that the optimization with *GradientMix* could reduce the trace of generalized Gauss-Newton matrix of the objective. Then, we provide the convergence analysis of the optimization with *GradientMix* in a non-convex regime under the standard assumptions.
- We validate the *GradientMix* popular problems in deep learning communities. Our extensive experiments show that *GradientMix* could result in better generalization especially for the large batch settings. Specifically, various optimizers with *GradientMix* consistently achieve state-of-the-art performance on the task of training ResNet-50 on the ImageNet with *32K batch size*.

## 2 METHOD

In this section, we introduce our main regularization technique, *GradientMix*. Then, we show that *GradientMix* encourages reducing the trace of generalized Gauss-Newton (GGN) of the loss function.

### 2.1 GRADIENTMIX: MIXING GRADIENTS WITH RANDOM MIXING RATE

As a motivating example, we start from the distributed training with multiple GPU machines. Let $\mathcal{D}$ be the training dataset with $n$ inputs $\mathcal{X} = \{x : (x, y) \in \mathcal{D}\}$ and the corresponding labels $\mathcal{Y} = \{y : (x, y) \in \mathcal{D}\}$. We study solving the optimization problem under the distributed environment:

$$\underset{\theta \in \mathbb{R}^d}{\text{minimize}} \ \mathcal{L}(\theta) := \frac{1}{n} \sum_{i=1}^{n} \ell(f(x_i; \theta), y_i) \tag{1}$$

where $\theta \in \mathbb{R}^d$ is the model parameter and $\ell(\widehat{y}, y)$ denotes the instance-wise loss function between the prediction $\widehat{y}$ and the true label $y$. Given $K$ devices, the vanilla SGD updates the model parameter $\theta$ as

$$g_{\text{final}} = \frac{1}{K} \sum_{k=1}^{K} \underbrace{\left[ \frac{1}{B} \sum_{b=1}^{B} \nabla \ell_{k_b}(f(\theta)) \right]}_{\text{local gradient}}, \quad \theta_{t+1} = \theta_t - \eta_t g_{\text{final}} \tag{2}$$

---

**Algorithm 1** A general optimization framework with *GradientMix*

---

1: **Input:** Training dataset $\mathcal{D}$, stepsize $\{\eta_t\}_{t=1}^T$, variance $\kappa > 0$, noise distribution $\mathbb{P}(\boldsymbol{\pi})$ satisfying Eq. 12, and optimization algorithm $\mathcal{A}$ (e.g. SGD, Adam, ...), momentum parameter $\mu \in [0, 1)$.
2: **Initialize:** Model parameter $\theta_1 \in \mathbb{R}^d$, (Nesterov) momentum $v_1 \in \mathbb{R}^d$.
3: **for** $t = 1, 2, \ldots, T$ **do**
4:     $\theta_{t+\frac{1}{2}} \leftarrow \theta_t + \mu v_t$
5:     **for** $b = 1, 2, \cdots, B$ **do**
6:         Randomly sample a $b$-th datapoint $x_t^{(b)}$ from $\mathcal{D}$     ▷ This can be done in a parallel way
7:         $g_t^{(b)} \leftarrow \nabla_\theta \mathcal{L}(x_t^{(b)}; \theta_{t+\frac{1}{2}})$     ▷ Sample-wise gradient
8:     **end for**
9:     Sampling random noise from noise distribution, $(\pi_1, \pi_2, \cdots, \pi_B) \sim \mathbb{P}(\boldsymbol{\pi})$
10:     $g_t \leftarrow \sum_{b=1}^B \pi_b g_t^{(b)}$     ▷ *GradientMix* in a sample-wise manner
11:     $v_{t+1} \leftarrow \mu v_t - \eta_t \sum_{b=1}^B \pi_t^{(b)} \nabla f_b(\theta_{t+\frac{1}{2}})$     ▷ Nesterov momentum construction
12:     $\theta_{t+1} \leftarrow \theta_t - \mathcal{A}(g_1, g_2, \cdots, g_t; \eta_t)$     ▷ Parameter update
13: **end for**
14: **Output:** $\theta_{T+1}$

---

where $B$ is the local batch size for each device and $k_b$ represents the $b$-th datapoint at device $k$ (hence, $b \in [B]$). For convenience, we abbreviate the gradient $\nabla \ell(f(x_i; \theta), y_i)$ evaluated at the datapoint $(x_i, y_i)$ as $\nabla \ell_i(f(\theta))$. In Eq. 2, the local gradient computed at device $k$ are averaged across all devices to yield the final gradient. Inspired by several studies (Srivastava et al., 2014; Zhu et al., 2019; Simsekli et al., 2019; Smith et al., 2020) on the importance of noise on generalization, we propose *mixing up* local gradients at each device $k$. Toward this, we first generate the random noise $\pi_k$ from some distribution $\mathbb{P}(\pi)$ at each device $k$ every iteration and perform linear combination of local gradients using $\pi_k$ as a mixing coefficient. In summary, the final gradient is computed by

$$\pi_k \sim \mathbb{P}(\pi) \text{ with } \mathbb{E}[\pi_k] = 1/K \quad \forall k \in [K], \quad g_{\text{final}} = \sum_{k=1}^K \frac{\pi_k}{B} \sum_{b=1}^B \nabla \ell_{k_b}(f(\theta)) \tag{3}$$

Preserving the unbiasedness of the stochastic gradient, we further assume that each noise $\pi_k$ satisfies $\mathbb{E}[\pi_k] = 1/K$ in Eq. 3, for example, Gaussian distribution with mean $1/K$.

In the one hand, the gradient mixing in Eq. 3 could also be applied in a sample-wise manner. Suppose that the gradient is computed with the batch size $B$, then the sample-wise gradient mixing would be

$$\pi_b \sim \mathbb{P}(\pi) \text{ with } \mathbb{E}[\pi_b] = 1/B \quad \forall b \in [B], \quad g_{\text{final}} = \sum_{b=1}^B \pi_b \nabla \ell_b(f(\theta)) \tag{4}$$

where $\pi_b$ is a sample-wise noise. The mixing gradients in a sample-wise manner Eq. 4 is a *generalized* extension of the batch-wise version in Eq. 3. We call the procedures in Eq. 4 as *GradientMix*.

Note that *GradientMix* is *optimizer-agnostic* since it does not require any specific update rule; therefore, it can be applicable to any popular optimizers such as Adam, LARS, LAMB, and etc. Thus, we consider a unified framework for *GradientMix*. We let $\mathcal{A}$ be a randomized optimization algorithm (e.g. SGD) and $\mathcal{A}(g_1, g_2, \cdots, g_t)$ produce the update vector at time $t$ where $g_\tau$ means the gradient computed at time $\tau$. Algorithm 1 summarizes an optimization framework with *GradientMix*.

One of the important properties of *GradientMix* is that the variance of mixed gradient is always higher than the traditional averaged gradient since the variance of mixed gradient in Eq. 4 is *smallest* when the noise $\pi_b$ in Eq. 4 satisfies $\pi_b = 1/B$ for all $b \in [B]$. We will investigate how this noisy property affect the loss landscape and the convergence of optimization in the following section.

## 2.2 GRADIENTMIX CAN REDUCE THE TRACE OF GENERALIZED GAUSS-NEWTON MATRIX

In this section, we provide more intuition of *GradientMix*. To this end, we consider an $L$-layer deep neural network following the notations in (Lee et al., 2019) defined by

$$h^{(l+1)} = x^{(l)} W^{(l+1)} + b^{(l+1)}, \quad x^{(l+1)} = \phi(h^{(l+1)})$$

for $x^{(0)} = x$ and $l = 0, 1, \cdots, L$. Here, $\phi(\cdot)$ is an element-wise (non-linear) activation function such as ReLU, $W^{(l)}$ and $b^{(l)}$ are the weight and bias parameter of appropriate shape at $l$-th layer, respectively. For simplicity, let $\theta \in \mathbb{R}^d$ be the vector of all network parameters and $f(x; \theta) = h^{(L+1)}(x) \in \mathbb{R}^k$ denote the network output (or logit). Under this setup, we first introduce the generalized Gauss-Newton (GGN) matrix. Toward this, we revisit an optimization problem in Eq. 1:

$$\underset{\theta}{\text{minimize}} \ \mathcal{L}(\theta) := \frac{1}{n} \sum_{i=1}^{n} \ell(f(x_i; \theta), y_i) = \frac{1}{n} \sum_{i=1}^{n} \ell_i(f(\theta)).$$

Thanks to (Schraudolph, 2002; Kunstner et al., 2019), the Hessian of the objective (with respect to $\theta$) can be written as

$$H(\theta) := \nabla_\theta^2 \mathcal{L}(\theta) = \frac{1}{n} \underbrace{\sum_{i=1}^{n} \widehat{\mathbf{J}}_\theta(x_i)^\mathsf{T} \nabla_f^2 \ell_i(f(\theta)) \widehat{\mathbf{J}}_\theta(x_i)}_{G(\theta)} + \text{Remaining term} \tag{5}$$

where $\widehat{\mathbf{J}}_\theta(x_i)$ represents the sample-wise Jacobian $\frac{\partial f}{\partial \theta}$ evaluated at single datapoint $(x_i, y_i)$ and $\nabla_f^2 \ell_i(f(\theta))$ is the Hessian of $\ell_i$ with respect to the network output. The matrix $G(\theta)$ defined in Eq. 5 is known to be the *generalized Gauss-Newton* (GGN) matrix. In case of convex loss $\ell$, the GGN is always positive (semi)definite and well-justified approximation of the Hessian $H(\theta)$ Kunstner et al. (2019), so it is a popular curvature approximation in the non-convex problems.

We study the relationship between the *GradientMix* and the GGN matrix. The gradient of objective function can be written as

$$\nabla_\theta \mathcal{L}(\theta) = \frac{1}{n} \sum_{i=1}^{n} \nabla_\theta \ell_i(f(\theta)) = \frac{1}{n} \sum_{i=1}^{n} \widehat{\mathbf{J}}_\theta(x_i)^\mathsf{T} \nabla_f \ell_i(f(\theta)) = Az \tag{6}$$

where $A$ is the matrix of all sample-wise gradients $A = [\nabla_\theta \ell_1(f(\theta)) \mid \cdots \mid \nabla_\theta \ell_n(f(\theta))] \in \mathbb{R}^{d \times n}$ and $z = (1/n, \cdots, 1/n) = \frac{1}{n}\mathbf{1} \in \mathbb{R}^n$ is a vector of $\frac{1}{n}$'s. Under this formulation, *GradientMix* in effect replaces the deterministic vector $z$ with the noisy one $\widetilde{z}$ as

$$\nabla_\theta \widetilde{\mathcal{L}}(\theta) = A\widetilde{z} \tag{7}$$

where the vector $\widetilde{z}$ is a random variable satisfying $\mathbb{E}[\widetilde{z}] = \frac{1}{n}\mathbf{1}$ for the unbiasedness of the gradient estimator as in Eq. 3. We call Eq. 7 a *mixed gradient*.

In the optimization analysis, the convergence is generally measured by *how fast the gradient norm* $\|\nabla_\theta \mathcal{L}(\theta)\|_2$ *approaches to zero*. *GradientMix* would change this convergence criterion to

$$\|\nabla_\theta \widetilde{\mathcal{L}}(\theta)\|_2 \to 0. \tag{8}$$

Plugging the mixed gradient Eq. 7 into Eq. 8, the convergence measure Eq. 8 is nothing but

$$\mathbb{E}[\widetilde{z}^\mathsf{T} A^\mathsf{T} A \widetilde{z}] \to 0 \tag{9}$$

Provided that $\widetilde{z}$ satisfies $\text{Cov}[\widetilde{z}] = \kappa^2 I$ for some positive constant $\kappa > 0$, owing to Hutchinson's trace estimator Hutchinson (1989), Eq. 9 is equivalent to

$$\text{Tr}(A^\mathsf{T} A) \to 0 \tag{10}$$

Rearranging the matrix in the trace operator, we finally arrive at the following:

$$\text{Tr}\left( \sum_{i=1}^{n} \widehat{\mathbf{J}}_\theta(x_i)^\mathsf{T} \nabla_f \ell_i(f(\theta)) \nabla_f \ell_i(f(\theta))^\mathsf{T} \widehat{\mathbf{J}}_\theta(x_i) \right) \to 0 \tag{11}$$

In case of a negative log-likelihood loss $\ell$ (as is many cases in deep learning problems), the matrix in the trace in equation Eq. 11 is the GGN matrix $G(\theta)$ (refer to Proposition 1 in Kunstner et al. (2019)). Consequently, the convergence with *GradientMix* roughly boils down to reducing the trace of GGN matrix, which could represents the shaprness of minima to some extent as in several previous work (Zhu et al., 2019; Simsekli et al., 2019; Lin et al., 2020). Hence, it might be possible that *GradientMix* could help to find the flatter solution than the one obtained via usual averaged gradient.

Note that this equivalence does not rely on the batch size, but we expect that *GradientMix* would be most helpful in a large batch regime, which is crucially vulnerable to the sharpness of the curvature.

The remaining question is that which noise class can reduce the trace of GGN matrix and whether the gradient norm really goes to zero when the optimization is equipped with *GradientMix*. For the first question, we already assume the mild conditions on the noise given the batch size $B$:

$$\mathbb{E}[\boldsymbol{\pi}] = \frac{1}{B}\mathbf{1}_B \text{ and } \mathrm{Cov}[\boldsymbol{\pi}] = \kappa^2 I_B \tag{12}$$

where $\mathbf{1}_B \in \mathbb{R}^B$ is just a vector of ones. There are several distributions that satisfy Eq. 12, and we adopt two practical distributions: (i) Gaussian and (ii) Rademacher.

Based on our mathematical intuition, we give the empirical evidence on the trace of GGN matrix. We train ResNet-18 on CIFAR-10 dataset with 8K batch size for large batch setting. The table 1 report the trace of GGN matrix of the final models trained with each method. As expected, *GradientMix* indeed reduces the trace of GGN to a meaningful extent, which means that *GradientMix* could find a flatter solution than the usual average approach with almost no additional computation overhead.

Table 1: The trace comparison for trained ResNet-18 on CIFAR-10 dataset.

| Method | $\|\log(\mathrm{Tr}(G(\theta)))$ |
|---|---|
| SGD Baseline | $\|$ 5.87 |
| SGD + *GradientMix* (Gaussian) | **4.21** |
| SGD + *GradientMix* (Rademacher) | **4.58** |

We show the effectiveness of *GradientMix* with these noise structures in large batch training in the experimental section. Now, we answer the second question in the next section.

## 3 CONVERGENCE ANALYSIS

In this section, we provide the convergence analysis of optimization with *GradientMix* in Algorithm 1. Our goal is to find an $\epsilon$-stationary point for the problem Eq. 1. For this purpose, we need the standard assumptions for the analysis in non-convex optimization:

**(C-1)** ($L$-smoothness) The loss function $f$ is differentiable, $L$-smooth, and lower-bounded: $\forall x, y, \ \|\nabla f(x) - \nabla f(y)\| \le L\|x - y\|$ and $f(x^*) > -\infty$ for the optimal solution $x^*$.

**(C-2)** (Bounded variance) The stochastic gradient $g_t$ at time $t$ is unbiased and has the bounded variance: $\mathbb{E}_\xi[g_t] = \nabla f(\theta_t), \quad \mathbb{E}_\xi[\|g_t - \nabla f(\theta_t)\|^2] \le \sigma^2$ where $\xi$ represents the randomness from data distribution.

The conditions (C-1) and (C-2) are standard in the analysis of non-convex optimization Ghadimi & Lan (2013); Zaheer et al. (2018); Chen et al. (2019); Yun et al. (2022). Also, *GradientMix* preserves the unbiasedness of the stochastic gradient, thereby satisfying the condition (C-2). Since Algorithm 1 presents the general framework with *GradientMix*, we specifically provide the convergence analysis for vanilla SGD with Nesterov momentum, which is the state-of-the-art optimizer used in deep learning communities. The update rule for SGD + Nesterov momentum with *GradientMix* would be

$$\theta_{t+\frac{1}{2}} = \theta_t + \mu v_t, \quad v_{t+1} = \mu v_t - \eta \sum_{b=1}^{B} \pi_t^{(b)} \nabla f_b(\theta_{t+\frac{1}{2}}), \quad \theta_{t+1} = \theta_t + v_{t+1} \tag{13}$$

where $\mu$ denotes the momentum parameter, $\pi_t^{(b)}$ is the sampled noise of $b$-th datapoint at time $t$ satisfying Eq. 12. We further define the quantities $\gamma_t$ and $\bar{\gamma}$ as

$$\gamma_t = B \sum_{b=1}^{B} (\pi_t^{(b)})^2, \quad \bar{\gamma} = \frac{1}{T} \sum_{t=0}^{T-1} \gamma_t$$

where $T$ is the total iterations. Note that the quantity $\gamma_t$ roughly measures the total amount of noise injected to the gradient at time $t$ and $\bar{\gamma}$ means the average amount of noise over all time. Now, we are ready to state our main theorem.

**Theorem 1** (Convergence for SGD + Nesterov momentum with *GradientMix*). *Let $\theta_a$ denote an iterate uniformly randomly chosen from $\{\theta_{\frac{1}{2}}, \cdots, \theta_{T+\frac{1}{2}}\}$. Under the conditions (C-1) and (C-2) with*

*the stepsize $\eta \leq \frac{2(1-\mu)^2}{L(\mu^3+1)}$, the Algorithm 1 with SGD + Nesterov momentum yields*

$$\mathbb{E}_a\Big[\|\nabla f(\theta_a)\|^2\Big] \leq \mathcal{O}\left(\frac{L\Delta(\mu^3+1)}{T(1-\mu)} + \sqrt{\frac{2L\Delta\sigma^2\bar{\gamma}}{BT(1-\mu)}}\right) \tag{14}$$

*where $\Delta = f(\theta_0) - f(\theta^*)$ with optimal point $\theta^*$.*

**Remarks.** Our Theorem 1 is a generalized version of Theorem 4.2 in Lin et al. (2020). Importantly, the second term in the upper bound in Eq. 14 is asymptotically dominant under the batch size condition $B = \mathcal{O}\left(\frac{1-\mu}{(\mu^3+1)^2} \times \frac{\sigma^2 T\bar{\gamma}}{L\Delta}\right)$. In this regime, we can obtain a linear speedup as increasing the batch size with the total iteration of order $\mathcal{O}\left(\frac{L\sigma^2\Delta\bar{\gamma}}{B\epsilon^2}\right)$ for $\epsilon$-stationary point. Note that the second term in the batch size condition involves the average noise $\bar{\gamma}$ attaining the smallest value 1 (so, $\bar{\gamma} \geq 1$) when the sampled probability is $\pi_t^{(b)} = 1/B$ for all $b$ and $t$, which corresponds to the traditional averaged gradients. In other words, the mixed gradient with random noise $\pi_t^{(b)}$ has always higher value of $\bar{\gamma}$ than the averaged gradient, which in effect allows larger critical batch size but with requiring more iterations. For this reason, *GradientMix* might show performance degradation in a tight budget of epochs, but tuning the variance $\kappa$ in Eq. 12, which affects $\bar{\gamma}$ in result, can play a role to balance the batch size and the number of total iterations.

Since *GradientMix* do not hurt the unbiasedness of the stochastic gradient but with a little higher variance, one can easily show the convergence of each optimizer with *GradientMix* upon the previous results. We defer the convergence analysis on LARS and LAMB with *GradientMix* in the Appendix.

## 4 RELATED WORK

**Large batch training.** Many researchers have paid attention to large-scale training to speed up training and achieve as high accuracy as small minibatch training. To reduce the generalization gap, Goyal et al. (2017) propose a linear scaling rule to increase the learning rate proportionally to the minibatch size. As exploiting a large learning rate can cause a neural network to diverge, You et al. (2017) suggest Layer-wise Adaptive Rate Scaling (LARS) to scale updates by multiplying the ratio of the $\ell_2$-norm of weights to that of gradients for each layer. Furthermore, You et al. (2019) apply this strategy to the Adam optimizer (Kingma & Ba, 2015), which is called LAMB, so as to accelerate training attention-based models, especially BERT, by using large batch sizes. In contrast to (You et al., 2017; 2019), without any layer-wise normalization, Nado et al. (2021) show that standard optimizers are enough for large batch training given sufficient hyperparmeter tuning. Apart from them, in order to narrow the generalization gap, Lin et al. (2020) employ an extragradient technique for smoothing and stabilizing optimization dynamics, and Johnson et al. (2020) propose AdaScale SGD that reliably and automatically adjusts the learning rate depending on the variance of gradients. However, all these methods mentioned above are based on the average of gradients, which is likely to preclude the possibility of exploration during training.

**Mixup.** Mixup Zhang et al. (2018) is one of the popular methods for enhancing the generalization and robustness of a neural network. Taking advantage of the beta distribution, Zhang et al. (2018) interpolates two inputs and their corresponding labels linearly to generate new data, and Verma et al. (2019) interpolates hidden representations linearly for a smooth decision boundary. Following these works, several extensions Yun et al. (2019); Kim et al. (2020) have been proposed to substitute some area of input for another input's patch, but all the aforementioned studies are discussed only in the context of data augmentation.

**Benefit of noise for network training.** In recent years, there have been several attempts to figure out connections between noise and generalization. Zhu et al. (2019) showed that noise in stochastic gradients helps escape from sharp minima and converge to flat minima which generalize well. Simsekli et al. (2019) analyze the dynamics of stochastic gradient descent driven by noise and showed that stochatic gradient descent prefers wide minima that performs better than narrow minima on the test set. Smith et al. (2020) verify that the noise in the stochastic gradient can improve generalization. Wu et al. (2020) showed that injecting multiplicative noise to the full-batch gradient generalizes as the usual stochastic gradient evaluated on mini-batch, but the noise structure they consider is not applicable to reducing the trace of curvature as we discussed in Section 2.2. Also, the effect of noise on the generalization is guaranteed in theory only in terms of the linear regression.

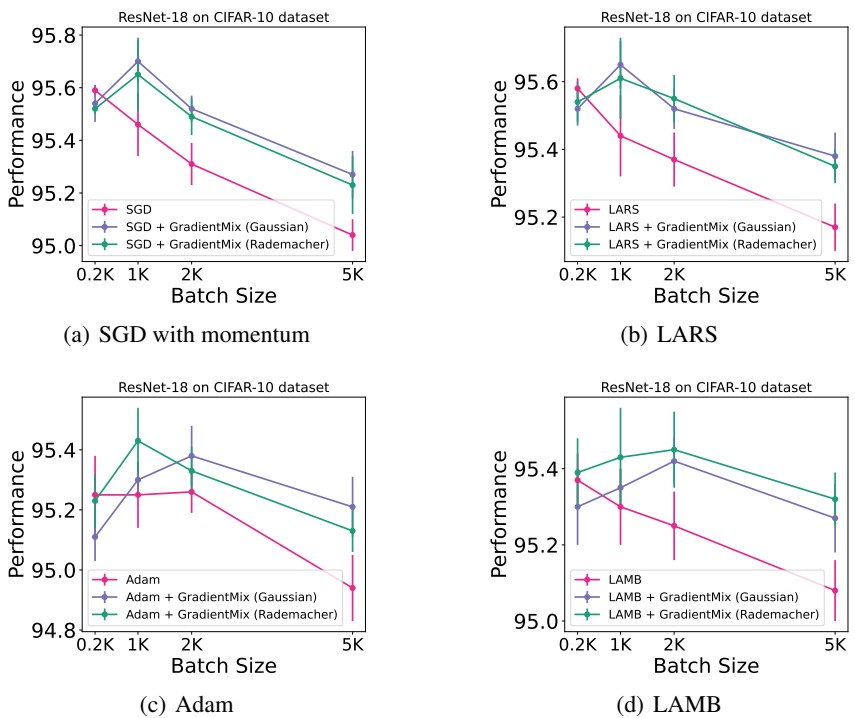

Figure 1: Results on training ResNet-18 on the CIFAR-10 dataset with *GradientMix*.

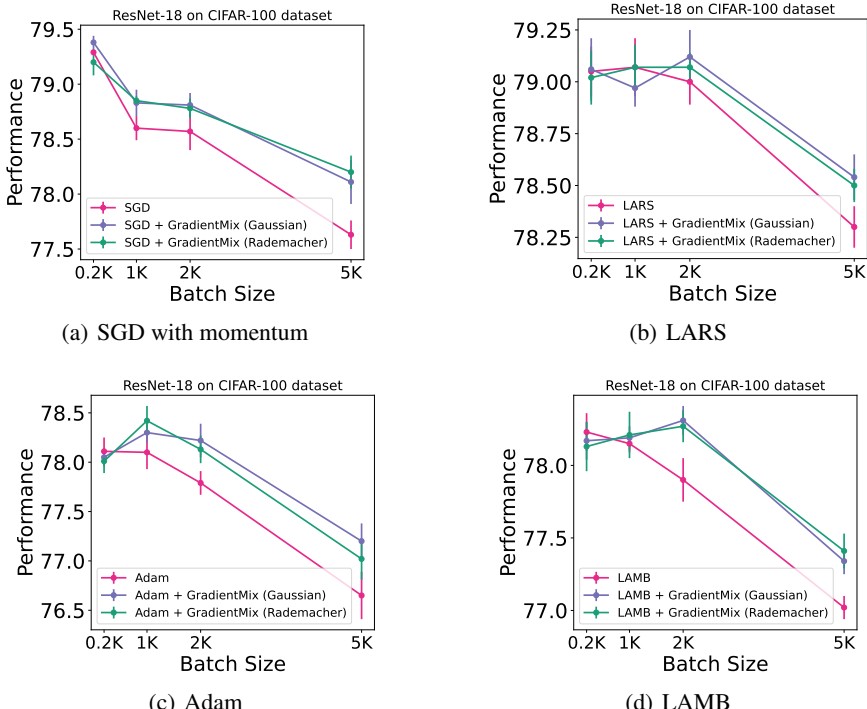

Figure 2: Results on training ResNet-18 on the CIFAR-100 dataset with *GradientMix*.

For more exploration in large batch training, instead of the averaged gradients, our work is the first in-depth study applying mixup to the local gradients using suitable distribution and we empirically justify that injecting appropriate noise into the gradient can bridge the generalization gap with a large batch as well in the next section.

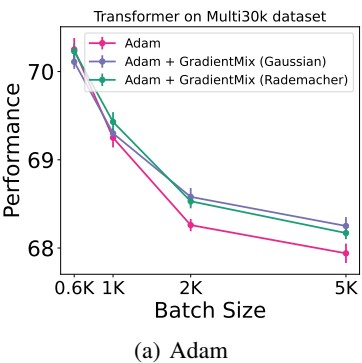 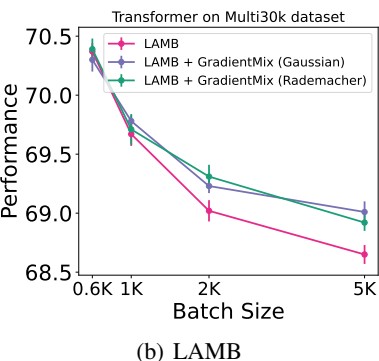

(a) Adam

(b) LAMB

Figure 3: Results on training Transformer on Multi30k dataset with *GradientMix*.

## 5 EXPERIMENTS

We consider two sets of experiments. The first set aims to purely see the effectiveness of *Gradient-Mix* on relatively small-scale problems and the second set is to evaluate *GradientMix* in a large scale. The details on experimental settings are provided at each section.

### 5.1 CIFAR CLASSIFICATION

We train ResNet-18 He et al. (2016) on the CIFAR datasets using two sets of optimizers, which is one of benchmark tasks in large batch training. The first set compares SGD with momentum and LARS, and the second set compares Adam and LAMB since LARS and LAMB are extensions of SGD and Adam respectively. For large batch training, the polynomial LR scheduling with gradual warmup Nado et al. (2021) is recommended, so we follow the linear LR scaling Goyal et al. (2017) with the base LR $\eta_{\text{base}} = 0.1$ for the batch size 200 for SGD and LARS. Similarly, we choose $\eta_{\text{base}} = 10^{-3}$ for the batch size 200 for Adam and LAMB. Throughout this experiment, we consider the total 200 epochs with 20 warmup epochs. In particular, the trust coefficient of LARS is fixed as $10^{-3}$. While our main goal is *how much GradientMix can improve the performance in a large batch regime*, we also include the results on the small batch size for better understanding.

Figure 1 illustrates the results for the CIFAR-10 dataset. *GradientMix* shows the consistent improvement in generalization for all optimizers considered acrosss all batch sizes except the smallest batch size 0.2K. The interesting point in Figure 1 is that for all optimizers, the performance with *GradientMix* tends to increase slightly and then decrease (see 0.2K $\sim$ 2K at $x$-axis) as the batch size increases while the generalization of all baselines only gets worse as the batch size gets larger. At *large batch size* such as 2K or 5K, which is *our main focus*, *GradientMix* show consistent superiority to the baselines without overlapping error bars. This might be due to the fact that the small batch size has already enough noise, in which *GradientMix* can interfere with model generalization.

The similar dynamics can be seen in Figure 2 for the CIFAR-100 dataset. Note that all the optimizers with *GradientMix* achieve a great improvement over the baseline for the largest batch size 5K as experiments for the CIFAR-10 dataset. Also, we can see the increase-then-decrease behavior in Figure 2 (see 0.2K $\sim$ 2K at $x$-axis) for all optimizers considered as in the CIFAR-10 experiment.

### 5.2 TRANSFORMER ON MULTI30K

In order to evaluate *GradientMix* on various problems in deep learning, we consider language modeling task. To this end, we train Transformer base model (Vaswani et al., 2017) on Multi30k dataset (Elliott et al., 2016). Following the experimental settings in Lin et al. (2020), we employ the linear LR scaling and gradual warmup scheme (Goyal et al., 2017) with inverse square root scheduling (Vaswani et al., 2017). The warmup step is set to 4000 for the base batch size 64 and linearly decayed by the batch size. It is known that the adaptive gradient methods work well for attention models, so we compare the Adam and LAMB optimizers for this experiment.

Figure 3 demonstrate the comparison of validation accuracy. Similar to the results in Section 5.1, *GradientMix* tends to be more effective as the batch size increases for both optimizers Adam and LAMB. Importantly, *GradientMix* achieve great improvements especially at large batch size such as

Table 2: Top-1 accuracy with *GradientMix* using *32K batch size* for training ResNet-50 on ImageNet.

| | Top-1 Accuracy (%) for 32K Batch Size | | | |
|---|---|---|---|---|
| | Optimization Algorithm | | | |
| | Fine-tuned hyperparameters with 64 epochs | | Standard hyperparameters with 90 epochs | |
| Noise Type | Nesterov | LARS | Adam | LAMB |
| No Noise (Baseline) | 75.93 ± 0.03 | 76.15 ± 0.04 | 76.13 ± 0.09 | 76.35 ± 0.08 |
| Gaussian (Ours) | **76.17** ± 0.06 | **76.30** ± 0.05 | **76.33** ± 0.06 | **76.50** ± 0.08 |
| Rademacher (Ours) | **76.25** ± 0.04 | **76.36** ± 0.07 | **76.31** ± 0.10 | **76.47** ± 0.07 |

2K and 5K. Combined with the results in Section 5.1, we believe that *GradientMix* has the potential to generalize better for other several tasks and optimizers in deep learning.

## 5.3 RESNET-50 ON IMAGENET WITH 32K BATCH SIZE

Training ResNet-50 on ImageNet has been considered as standard benchmark in large batch optimization, so we evaluate our *GradientMix* on this task using various optimization algorithms. In large batch training, several regularization techniques and hyperparameters should be considered carefully in order to achieve the comparable performance of small batch training. Toward this, we first introduce the experiment settings for each optimizer.

The recent study Nado et al. (2021) empirically shows that the traditional optimizers such as Nesterov can achieve the competitive performance to the optimizer specifically designed for large batch training, such as LARS or LAMB, with the same effort in the hyperparameter tuning. Inspired by this work, we investigate whether *GradientMix* can further enhance the generalization even under highly fine-tuned hyperparameters. Recommended in Nado et al. (2021), the weight decay should be applied only to the network weight parameters except the bias and batch normalization parameters (You et al., 2017; Goyal et al., 2017). One of the most essential tricks to improve the generalization (He et al., 2019b; Nado et al., 2021) is to tune the initial scale parameter $\gamma_0$ of the *final batch normalization layer of each residual block* (Goyal et al., 2017). We use the polynomial LR scheduling (Nado et al., 2021) with tuning each hyperparameter. We summarize the hyperparameter details in the Appendix.

For Adam and LAMB, unfortunately, there is no previous reference using highly fine-tuned hyperparamaters for ResNet-50 training, so we follow the experimental pipelines in (You et al., 2019). We use the momentum parameters $(\beta_1, \beta_2) = (0.9, 0.999)$ with $\epsilon = 10^{-6}$ for both Adam and LAMB optimizers as in (You et al., 2019) and the polynomial LR scheduling with gradual warmup (Nado et al., 2021) is employed. The number of warmup epochs $t_{\text{warmup}}$ is set to 20 among total 90 epochs.

Table 2 illustrates the Top-1 accuracy of each method for 32K batch size. For all baseline optimizers, *GradientMix* could successfully achieve the state-of-the-art generalization even under highly fine-tuned hyperparameters (see Nesterov and LARS) as well as under standard hyperparameter settings (see Adam and LAMB). As seen in Table 2, the performance gain for *GradientMix* with Nesterov and Adam is slightly larger than that for *GradientMix* with LARS and LAMB respectively. This might be due to the fact that LARS and LAMB algorithms are already optimized designs to some extent specifically for large batch training, but our results prove that *GradientMix* is indeed optimizer-agnostic. We emphasize that our improvement (without overlapping confidence intervals) is NOT marginal since the performance of baselines is already highly optimized for ResNet-50 architecture. Also, our improvements are bigger than the ones achieved in recent studies related to regularization in a large batch regime such as Yuan et al. (2020); Huo et al. (2021).

## 6 CONCLUSION

We proposed an optimizer-agnostic and effective regularization for large-batch training, *GradientMix*, which is the mixup of local gradients with arbitrarily sampled noise. We also show that *GradientMix* could reduce the sharpness of the loss landscape. Finally, we empirically verify the effectiveness of *GradientMix* on various datasets/models and achieve state-of-the-art performance on the benchmark task, training ResNet-50 on ImageNet dataset. As future work, we plan to investigate how the various type of noise affects the performance for large-batch training both in theory and practice.

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

## SUPPLEMENTARY MATERIALS

## A HYPERPARAMETER DETAILS FOR NESTEROV AND LARS IN SECTION 5.3

As described in Section 5.3, we follow the same hyperparameter settings in Nado et al. (2021) for Nesterov and LARS optimizers. First, we employ the polynomial learning rate scheduling designed for large-batch training:

$$\eta_t = \begin{cases} \eta_{\text{init}} + (\eta_{\text{peak}} - \eta_{\text{init}}) \left( \frac{t}{t_{\text{warmup}}} \right)^{p_{\text{warmup}}}, & t \leq t_{\text{warmup}} \\ \eta_{\text{final}} + (\eta_{\text{peak}} - \eta_{\text{final}}) \left( \frac{T-t}{T-t_{\text{warmup}}} \right)^{p_{\text{decay}}}, & t > t_{\text{warmup}} \end{cases} \tag{15}$$

We train ResNet-50 on ImageNet dataset with the batch size $B = 32768$ (32K). We summarize the detailed values for each hyperparameter introduced in Section 5.3 in the following table.

Table 3: The highly fine-tuned hyperparameter configuration for training ResNet-50 on ImageNet dataset for Nesterov and LARS optimizers.

|  | Nesterov | LARS |
| --- | --- | --- |
| Total steps (Corresponding to $64$ epochs) | 2512 | 2512 |
| $t_{\text{warmup}}$ | 706 | 706 |
| $p_{\text{warmup}}$ | 2.0 | 1.0 |
| $p_{\text{decay}}$ | 2.0 | 2.0 |
| BatchNorm momentum | 0.9 | 0.9 |
| BatchNorm $\epsilon$ | $10^{-5}$ | $10^{-5}$ |
| $\eta_{\text{peak}}$ | 7.05 | 29.0 |
| $\eta_{\text{final}}$ | $6 \times 10^{-6}$ | $10^{-4}$ |
| Optimizer momentum | 0.97603 | 0.929 |
| Weight decay | $5.8 \times 10^{-5}$ | $10^{-4}$ |
| Label smoothing | 0.15 | 0.1 |
| Initial scale parameter $\gamma_0$ of the final BN layer of each residual block | 0.4138 | 0.0 |
| Trust coefficient (only for LARS) | N/A | $10^{-3}$ |

## B PROOF OF THEOREM 1

Our analysis is based on the convergence proof of SGD + Nesterov momentum Lin et al. (2020). We first derive the convergence for batch-wise *GradientMix* and then present the analysis for sample-wise *GradientMix*. Recall that the iterate of SGD + Nesterov momentum with *GradientMix* is

$$\theta_{t+\frac{1}{2}} = \theta_t + \mu v_t, \quad v_{t+1} = \mu v_t - \frac{\eta}{B} \sum_{k=1}^{K} \pi_t^{(k)} \sum_{b=1}^{B} \nabla f_{k_b}(\theta_{t+\frac{1}{2}}), \quad \theta_{t+1} = \theta_t + v_{t+1}$$

where $\pi_t^{(k)}$ means that the sampled probability of $k$-th device at time $t$ and $\mu$ is the momentum parameter. For simplicity, we define the $g_t$ as

$$g_t = \frac{1}{B} \sum_{k=1}^{K} \pi_t^{(k)} \sum_{b=1}^{B} \nabla f_{k_b}(\theta_t)$$

Then, the momentum update would become

$$v_{t+1} = \mu v_t - \eta g_{t+\frac{1}{2}}$$

We also define the quantities $\zeta_t$ and $\bar{\zeta}$ as

$$\zeta_t = \sum_{k=1}^{K} (\pi_t^{(k)})^2, \quad \bar{\zeta} = \frac{1}{T} \sum_{t=0}^{T-1} \zeta_t$$

The main changes in our analysis is how to bound the term $\mathbb{E}\left[\frac{1}{T}\sum_{t=0}^{T-1}\|g_{t+\frac{1}{2}}\|^2\right]$. Toward this, we have

$$\mathbb{E}\left[\|g_{t+\frac{1}{2}}\|^2\right] = \mathbb{E}\left[\left\|\frac{1}{B}\sum_{k=1}^{K}\pi_t^{(k)}\sum_{b=1}^{B}\nabla f_{k_b}(\theta_{t+\frac{1}{2}})\right\|^2\right] \tag{16}$$

$$= \mathrm{Var}\left(\frac{1}{B}\sum_{k=1}^{K}\pi_t^{(k)}\sum_{b=1}^{B}\nabla f_{k_b}(\theta_{t+\frac{1}{2}})\right) + \left\|\mathbb{E}\left[\frac{1}{B}\sum_{k=1}^{K}\pi_t^{(k)}\sum_{b=1}^{B}\nabla f_{k_b}(\theta_{t+\frac{1}{2}})\right]\right\|^2 \tag{17}$$

$$= \frac{\zeta_t}{B}\sigma^2 + \left\|\nabla f(\theta_{t+\frac{1}{2}})\right\|^2 \tag{18}$$

Hence, we obtain

$$\mathbb{E}\left[\frac{1}{T}\sum_{t=0}^{T-1}\left\|g_{t+\frac{1}{2}}\right\|^2\right] \leq \frac{\bar{\zeta}}{B}\sigma^2 + \frac{1}{T}\sum_{t=0}^{T-1}\left\|\nabla f(\theta_{t+\frac{1}{2}})\right\|^2 \tag{19}$$

Following the proofs in Lin et al. (2020), we define an auxiliary sequence $y_t$ as

$$y_t = \begin{cases} x_{\frac{1}{2}} = x_0 & \text{if } t = 0 \\ \frac{1}{1-\mu}\theta_{t+\frac{1}{2}} - \frac{\mu}{1-\mu}\theta_{t-\frac{1}{2}} + \frac{\eta\mu}{1-\mu}g_{t-\frac{1}{2}} & \text{if } t \geq 1 \end{cases} \tag{20}$$

**Lemma 1** (Lemma A.1 in Lin et al. (2020)). *The sequence $\{y_t\}$ in equation 20 satisfies*

$$y_{t+1} - y_t = -\frac{\eta}{1-\eta}g_{t+\frac{1}{2}}$$

**Lemma 2** (Lemma A.2 in Lin et al. (2020)). *For a sequence $\{x_{t+\frac{1}{2}}\}$ for $t \geq 0$, the following holds*

$$\sum_{t=0}^{T-1}\left\|y_t - \theta_{t+\frac{1}{2}}\right\|^2 \leq \frac{\mu^4\eta^2}{(1-\mu)^4}\sum_{t=0}^{T-1}\left\|g_{t+\frac{1}{2}}\right\|^2$$

Now, we derive the convergence bound. From smoothness condition, we have

$$\mathbb{E}[f(y_{t+1}) - f(y_t)] \leq \mathbb{E}\left[\langle\nabla f(y_t), y_{t+1} - y_t\rangle + \frac{L}{2}\|y_{t+1} - y_t\|^2\right]$$

$$= \mathbb{E}\left[-\frac{\eta}{1-\mu}\left\|\nabla f(\theta_{t+\frac{1}{2}})\right\|^2 - \frac{\eta}{1-\mu}\left\langle\nabla f(y_t) - \nabla f(x_{t+\frac{1}{2}}), \nabla f(\theta_{t+\frac{1}{2}})\right\rangle + \frac{L}{2}\left\|\frac{\eta}{1-\mu}g_{t+\frac{1}{2}}\right\|^2\right]$$

by Lemma 1. Further we have

$$-\frac{\eta}{1-\mu}\left\langle\nabla f(\theta_t) - \nabla f(\theta_{t+\frac{1}{2}}), \nabla f(\theta_{t+\frac{1}{2}})\right\rangle = \left\langle-\frac{\sqrt{1-\mu}}{\sqrt{L}u^{3/2}}\left(\nabla f(y_t) - \nabla f(\theta_{t+\frac{1}{2}})\right), \frac{\eta\sqrt{L}u^{3/2}}{(1-\mu)^{3/2}}\nabla f(\theta_{t+\frac{1}{2}})\right\rangle$$

$$\leq \frac{1-\mu}{2Lu^3}\left\|\nabla f(y_t) - \nabla f(x_{t+\frac{1}{2}})\right\|^2 + \frac{\eta^2 L\mu^3}{2(1-\mu)^3}\left\|\nabla f(\theta_{t+\frac{1}{2}})\right\|^2$$

by the inequality $\langle x, y\rangle \leq \frac{1}{2}(\|x\|^2 + \|y\|^2)$. Then, we get

$$\mathbb{E}[f(y_{t+1}) - f(y_t)]$$

$$\leq \mathbb{E}\left[-\frac{\eta}{1-\mu}\left\|\nabla f(\theta_{t+\frac{1}{2}})\right\|^2 + \frac{1-\mu}{2L\mu^3}\left\|\nabla f(y_t) - \nabla f(\theta_{t+\frac{1}{2}})\right\|^2 + \frac{\eta^2 L\mu^3}{2(1-\mu)^3}\left\|\nabla f(\theta_{t+\frac{1}{2}})\right\|^2 + \frac{\eta^2 L}{2(1-\mu)^2}\left\|g_{t+\frac{1}{2}}\right\|^2\right]$$

$$\leq \mathbb{E}\left[\left(-\frac{\eta}{1-\mu} + \frac{\eta^2 L\mu^3}{2(1-\mu)^3}\right)\left\|\nabla f(\theta_{t+\frac{1}{2}})\right\|^2 + \frac{(1-\mu)L}{2\mu^3}\left\|y_t - \theta_{t+\frac{1}{2}}\right\|^2 + \frac{\eta^2 L}{2(1-\mu)^2}\left\|g_{t+\frac{1}{2}}\right\|^2\right]$$

By telescoping over $t = 0 \sim T - 1$, we obtain

$$\frac{1}{T}\sum_{t=0}^{T-1}\mathbb{E}[f(y_{t+1}) - f(y_t)] = \frac{1}{T}\mathbb{E}[f(y_t) - f(y_0)]$$

$$\leq \left(-\frac{\eta}{1-\mu} + \frac{L\eta^2\mu^3}{2(1-\mu)^3}\right)\frac{1}{T}\sum_{t=0}^{T-1}\left\|\nabla f(\theta_{t+\frac{1}{2}})\right\|^2 + \mathbb{E}\left[\frac{(1-\mu)L\eta^2}{2\mu^3}\frac{\mu^4}{(1-\mu)^4}\frac{1}{T}\sum_{t=0}^{T-1}\left\|g_{t+\frac{1}{2}}\right\|^2\right]$$

$$+ \mathbb{E}\left[\frac{\eta^2 L}{2(1-\mu)^2}\frac{1}{T}\sum_{t=0}^{T-1}\left\|g_{t+\frac{1}{2}}\right\|^2\right]$$

Here, we apply our derivations equation 18 then have

$$\mathbb{E}[f(y_t) - f(y_0)]$$
$$\leq \left(-\frac{\eta}{1-\mu} + \frac{L\eta^2\mu^3}{2(1-\mu)^3} + \frac{\eta^2 L}{2(1-\mu)^2} + \frac{L\mu\eta^2}{2(1-\mu)^3}\right)\frac{1}{T}\sum_{t=0}^{T-1}\left\|\nabla f(\theta_{t+\frac{1}{2}})\right\|^2 + \left(\frac{\eta^2 L}{2(1-\mu)^2} + \frac{L\mu\eta^2}{2(1-\mu)^3}\right)\frac{\bar{\zeta}\sigma^2}{B}$$

By rearranging all the items, we have

$$\frac{1}{T}\sum_{t=0}^{T-1}\left\|\nabla f(\theta_{t+\frac{1}{2}})\right\|^2 \leq \frac{1}{1 - \frac{L\eta(\mu^3+1)}{2(1-\mu)^2}}\left(\frac{1}{T\frac{\eta}{1-\mu}}\Delta + \frac{\eta L}{2(1-\mu)^2}\frac{\bar{\zeta}\sigma^2}{B}\right)$$

In order to get the final results, we need the following lemmas from Lin et al. (2020)

**Lemma 3** (Lemma A.4 in Lin et al. (2020))**.** *For every non-negative sequence $\{r_t\}_{t\geq 0}$ and any parameters $d \geq 0$, $c \geq 0$, and $T \geq 0$, there exists a constant $\eta \leq 1/d$ such that for any constant stepsizes $\eta_t = \eta$, it holds*

$$\Psi_T := \frac{1}{T+1}\sum_{t=0}^{T}\left(\frac{r_t}{\eta_t} - \frac{r_{t+1}}{\eta_t} + c\eta_t\right) \leq \frac{d\Delta}{\eta(T+1)} + c\eta \tag{21}$$

Then, using $\Psi'_T := \frac{\Delta}{T\frac{\eta}{1-\mu}} + \frac{\eta L}{2(1-\mu)^2}\frac{\bar{\zeta}\sigma^2}{B}$ in Lemma 3, we can arrive at the following results by case study where $\frac{\Delta B}{L\sigma^2\zeta T} \leq \frac{1}{L^2}$ or $> \frac{1}{L^2}$ similar to Lin et al. (2020),

$$\mathbb{E}\left[\frac{1}{T}\sum_{t=0}^{T-1}\left\|\nabla f(\theta_{t+\frac{1}{2}})\right\|^2\right] = \mathcal{O}\left(\frac{L\Delta(\mu^3+1)}{T(1-\mu)} + \sqrt{\frac{2L\Delta\sigma^2\bar{\zeta}}{BT(1-\mu)}}\right) \tag{22}$$

Now, if we replace the $K = B_0$ and $B = 1$ for some batch size $B_0$, we obtain the convergence of sample-wise *GradientMix*. Going further, for batch size $B_0$, the noise quantity $\zeta_t$ in equation 16 should be at least of order $1/B_0$, in other words

$$\zeta_t = \sum_{b=1}^{B_0}(\pi_t^{(b)})^2 \geq 1/B_0 = \Omega(1/B_0)$$

since $\zeta_t$ is smallest when all the noise satisfy $\pi_t^{(b)} = 1/B_0$ (by Cauchy-Schwarz inequality). For clear analysis, in order to remove the dependency on the batch size $B_0$, we newly define the following quantity

$$\gamma_t = B_0\sum_{b=1}^{B_0}(\pi_t^{(b)})^2, \quad \bar{\gamma} = \frac{1}{T}\sum_{t=0}^{T-1}\gamma_t$$

Then, the final convergence bound for Nesterov with sample-wise *GradientMix* would be

$$\mathbb{E}\left[\frac{1}{T}\sum_{t=0}^{T-1}\left\|\nabla f(\theta_{t+\frac{1}{2}})\right\|^2\right] = \mathcal{O}\left(\frac{L\Delta(\mu^3+1)}{T(1-\mu)} + \sqrt{\frac{2L\Delta\sigma^2\bar{\gamma}}{B_0 T(1-\mu)}}\right) \tag{23}$$

for the batch size $B_0$ (in effect, replaces $\bar{\zeta}$ with $\bar{\gamma}$).

## B.1 CONVERGENCE ANALYSIS OF LARS WITH *GradientMix*

From this section, we analyze the convergence of LARS/LAMB with gradient mixup based upon the previous analysis in You et al. (2019). Here, we make the conditions specific to LARS and LAMB as follows.

We assume that the loss function $f(\cdot)$ is $L_l$-smooth with respect to the paramter of the $l$-th layer $\theta^{(l)}$ for $l \in [h]$, which means

$$\|\nabla_l f(x, s) - \nabla_l f(y, s)\| \leq L_i\|x^{(l)} - y^{(l)}\|, \forall x, y \in \mathbb{R}^d$$

where $s$ represents a random datapoint from data distribution. We use $L = (L_1, \cdots, L_h)^\mathsf{T}$. We also assume the bounded variance of stochastic gradient as

$$\mathbb{E}[\|\nabla_l f(x, s) - \nabla_l f(x)\|^2] \leq \sigma_l^2, \forall x \in \mathbb{R}^d$$

for all layer $i \in [h]$. Furthermore, as in You et al. (2019), we assume that

$$\mathbb{E}[\|[\nabla f(x,s)]_j - [\nabla f(x)]_j\|^2] \leq \widetilde{\sigma}_i^2, \forall x \in \mathbb{R}^d$$

for $j \in [d]$. Under this condition, we use the following notation for simplicity as

$$\sigma = (\sigma_1, \cdots, \sigma_h)^\mathsf{T}, \quad \widetilde{\sigma} = (\widetilde{\sigma}_1, \cdots, \widetilde{\sigma}_d)^\mathsf{T}$$

Lastly, we assume the bounded gradient condition as $[\nabla f(x,s)]_j \leq G$ for all $j \in [d]$ and $x \in \mathbb{R}^d$.

Recall that the iterate of LARS You et al. (2019) is

$$\theta_{t+1}^{(l)} = \theta_t^{(l)} - \eta\phi(\|\theta_t^{(l)}\|)\frac{g_t^{(l)}}{\|g_t^{(l)}\|} \tag{24}$$

for all layer $l \in [L]$ and $\theta_t^{(l)}$ represents the parameter of $l$-th layer at time $t$. The scaling function $\phi(\cdot)$ satisfies $\alpha_l \leq \phi(\cdot) \leq \alpha_u$ and in practice we use just a clipping function $\phi(z) = \min\{\max\{z, \gamma_l\}, \gamma_u\}$. Here, the stochastic gradient $g_t$ with gradient mixup is computed as

$$g_t = \sum_{b=1}^{B} \pi_t^{(b)} \nabla f(\theta_t, s_b) \tag{25}$$

where $s_b$ denotes the $b$-th sample in a mini-batch of size $B$ and $\{\pi_t^{(b)}\}_{b=1}^{B}$ is a unit simplex at time $t$. Note that this is more general setting of gradient mixup and datapoint-wise gradient mixup can surely cover the case of minibatch-wise gradient mixup. Here, we define the following quantities $\zeta_t$ and $\bar{\zeta}$ similarly as Nesterov momentum

$$\zeta_t = \sum_{b=1}^{B} (\pi_t^{(b)})^2, \quad \bar{\zeta} = \frac{1}{T}\sum_{t=1}^{T} \sqrt{\zeta_t}$$

The main changes in our analysis is how to bound the term

$$\mathbb{E}\left[\|g_t^{(l)} - \nabla_i f(\theta_t)\|\right] \tag{26}$$

where $\nabla_i f(\theta_t)$ means the gradient of loss function computed at time $t$ with respect to the parameter $\theta_t^{(l)}$ of $l$-th layer. The above term can be easily bound under the bounded varaince condition as

$$\mathbb{E}\left[\|g_t^{(l)} - \nabla_i f(\theta_t)\|^2\right] = \mathbb{E}\left[\|\sum_{b=1}^{B} \pi_t^{(b)} \nabla_i f(\theta_t, s_b) - \nabla_i f(\theta_t)\|^2\right] \tag{27}$$

$$= \mathbb{E}\left[\|\sum_{b=1}^{B} \pi_t^{(b)}(\nabla_i f(\theta_t, s_b) - \nabla_i f(\theta_t))\|^2\right] \tag{28}$$

$$= \mathrm{Var}\left(\sum_{b=1}^{B} \pi_t^{(b)}(\nabla_i f(\theta_t, s_b) - \nabla_i f(\theta_t))\right) + \underbrace{\left\|\mathbb{E}\left[\sum_{b=1}^{B} \pi_t^{(b)}(\nabla_i f(\theta_t, s_b) - \nabla_i f(\theta_t))\right]\right\|^2}_{0} \tag{29}$$

$$= \zeta_t \sigma_i^2 \tag{30}$$

Hence, we have

$$\mathbb{E}\left[\|g_t^{(l)} - \nabla_i f(\theta_t)\|\right] \leq \sqrt{\zeta_t}\sigma_i \tag{31}$$

We replace the inequality (6) in the Appendix A in You et al. (2019) with our equation 31. Then, the following holds

$$\mathbb{E}[f(\theta_{t+1})] \leq f(\theta_t) - \eta\alpha_l \sum_{l=1}^{h} \|\nabla_l f(\theta_t)\| + 2\eta\alpha_u \sqrt{\zeta_t}\|\sigma\|_1 + \frac{\eta^2\alpha_u^2}{2}\|L\|_1$$

By telescoping over $t = 1 \sim T$, we have

$$\mathbb{E}[f(\theta_{T+1})] \leq f(\theta_1) - \eta\alpha_l \sum_{t=1}^{T}\sum_{l=1}^{h} \mathbb{E}[\nabla_l f(\theta_t)\|] + 2\eta T\alpha_u\bar{\zeta}\|\sigma\|_1 + \frac{\eta^2\alpha_u^2 T}{2}\|L\|_1$$

Now, we arrive at the final convergence rate as

$$\frac{1}{T}\sum_{t=1}^{T}\sum_{l=1}^{h} \mathbb{E}[\|\nabla_l f(\theta_t)\|] \leq \frac{\Delta}{T\eta\alpha_l} + \frac{2\alpha_u\bar{\zeta}\|\sigma\|_1}{\alpha_l} + \frac{\eta\alpha_u^2}{2\alpha_l}\|L\|_1$$

where $\Delta = f(\theta_1) - f(\theta^*)$ with the optimal point $\theta^*$.

## B.2 CONVERGENCE ANALYSIS OF LAMB WITH *GradientMix*

In this section, we provide the convergence analysis of LAMB, which has the following update rule

$$g_t = \sum_{b=1}^{B} \pi_t^{(b)} \nabla f(\theta_t, s_b)$$

$$m_t = \beta_1 m_{t-1} + (1 - \beta_1) g_t$$

$$v_t = \beta_2 v_{t-1} + (1 - \beta_2) g_t^2$$

$$m_t = m_t / (1 - \beta_1^t)$$

$$v_t = v_t / (1 - \beta_2^t)$$

$$r_t^{(l)} = \frac{m_t^{(l)}}{\sqrt{v_t^{(l)}} + \epsilon}$$

$$\theta_{t+1}^{(l)} = \theta_t^{(l)} - \eta_t \frac{\phi(\|\theta_t^{(l)}\|)}{\|r_t^{(l)} + \lambda \theta_t^{(l)}\|} (r_t^{(l)} + \lambda \theta_t^{(l)})$$

As in You et al. (2019), we deal with the case of $\beta_1 = 0$ and $\lambda = 0$, but $\beta_2 > 0$. Then, the update vector would be

$$r_t^{(l)} = \frac{g_t^{(l)}}{\sqrt{v_t^{(l)}} + \epsilon}$$

Similar to the convergence of LARS, the major changes in our analysis is in how to bound the following term from inequality (7) in the Appendix You et al. (2019)

$$T_1 \leq -\eta \sum_{l=1}^{h} \sum_{j=1}^{d_l} \sqrt{\frac{1-\beta_2}{G^2 d_l}} \left( \phi(\|\theta_t^{(l)}\|) \times [\nabla_l f(\theta_t)]_j \times g_{t,j}^{(l)} \right)$$

$$- \eta \sum_{l=1}^{h} \sum_{j=1}^{d_l} \left( \phi(\|\theta_t^{(l)}\| \times [\nabla_l f(\theta_t)]_j \times \frac{r_t^{(l)}}{\|r_t^{(l)}\|} \mathbb{K}(\text{sign}(\nabla_l f(\theta_t)]_j) \neq \text{sign}(r_{t,j}^{(l)})) \right)$$

$$\leq -\eta \sum_{l=1}^{h} \sum_{j=1}^{d_l} \sqrt{\frac{1-\beta_2}{G^2 d_l}} \mathbb{E} \left[ \phi(\|\theta_t\|) \times \left( [\nabla_l f(\theta_t)]_j \times g_{t,j}^{(l)} \right) \right]$$

$$- \eta \sum_{l=1}^{h} \sum_{j=1}^{d_l} \alpha_u \left| [\nabla_l f(\theta_t)]_j \right| \mathbb{P}(\text{sign}([\nabla_l f(\theta_t)]_j) \neq \text{sign}(g_{t,j}^{(l)}))$$

Here, the mixed gradient has a different bound with the averaged gradient as

$$\mathbb{E}[T_1] \leq -\eta \alpha_l \sqrt{\frac{L(1-\beta_2)}{G^2 d}} \|\nabla f(\theta_t)\|^2 + \eta \alpha_u \sqrt{\zeta_t} \sum_{l=1}^{h} \sum_{j=1}^{d_l} \sigma_{l,j}$$

Therefore, we have

$$\mathbb{E}[f(\theta_{t+1})] \leq f(\theta_t) - \eta \alpha_l \sqrt{\frac{L(1-\beta_2)}{G^2 d}} \|\nabla f(\theta_t)\|^2 + \eta \alpha_u \sqrt{\zeta_t} \|\tilde{\sigma}\|_1 + \frac{\eta^2 \alpha_u^2 \|L\|_1}{2}$$

Telescoping over $t = 1 \sim T$ yields

$$\mathbb{E}[f(\theta_{t+1})] \leq f(\theta_1) - \eta \alpha_l \sqrt{\frac{L(1-\beta_2)}{G^2 d}} \sum_{t=1}^{T} \mathbb{E}[\|\nabla f(\theta_t)\|^2] + \eta T \alpha_u \bar{\zeta} \|\tilde{\sigma}\|_1 + \frac{\eta^2 \alpha_u^2 T}{2} \|L\|_1$$

Finally, we have the following convergence guarantee of LAMB optimizer with gradient mixup as

$$\sqrt{\frac{L(1-\beta_2)}{G^2 d}} \frac{1}{T} \sum_{t=1}^{T} \mathbb{E}[\|\nabla f(\theta_t)\|^2] \leq \frac{\Delta}{T \eta \alpha_l} + \frac{\alpha_u \bar{\zeta} \|\tilde{\sigma}\|_1}{\alpha_l} + \frac{\eta \alpha_u^2}{2 \alpha_l} \|L\|_1$$

