# OpenReview forum: "GradientMix: A Simple yet Effective Regularization for Large Batch Training"
_ICLR.cc/2023/Conference — Submitted to ICLR 2023_

### Official Review · Reviewer_LFbe · 2022-10-24

**Confidence:** 3
**Clarity, Quality, Novelty And Reproducibility:** NA
**Correctness:** 3
**Technical Novelty And Significance:** 3
**Empirical Novelty And Significance:** 3
**Recommendation:** 5

**Strength And Weaknesses:**

Strength:
This paper is very easy to follow. The authors propose GeadientMix to solve the generalization issue in large-batch training.
The proposed method is easy to reproduce. As shown in Algorithm 1, the main difference is about sampling random noise and weighted gradient.


Weakness:

I still think the proposed method is not novel. The proposed method (mixup of local gradients computed at multiple devices) is similar to the weighted gradient.

The main purpose of Large-batch training is to accelerate the training process. Therefore, I think you also need to report the training time for these methods. Since GradientMix introduces more computation cost.

As you mentioned, generalization is an important issue for large-batch training. So I think you should also report the generalization gap (training accuracy - test accuracy) in your experiments to verify GradientMix can narrow the gap.

You need to provide more experiments about sensitivity analysis. For example, the results with different number of GPUs and different noise. That is because the number of GPUs and the selection of noise (with different mean and variance) is an important hyper-parameters for your proposed method.

More results about recent regularization methods for large-batch training should be reported. The main reason is that LARS  is proposed in 2017 and there are many algorithms that try to further improve the performance of large-batch training.

**Summary Of The Paper:**

This paper tries to use large-batch training to accelerate neural network training. However, traditional large-batch training algorithms usually suffer from generalization issues. To solve this problem, this paper proposes an algorithm GradientMix, which tries to mixup of local gradients computed at multiple devices. The experimental results illustrate that GradientMix can further improve the accuracy of LARS/LAMB for large-batch training.

**Summary Of The Review:**

NA

---

### Official Review · Reviewer_HmFS · 2022-10-29

**Confidence:** 4
**Correctness:** 4
**Technical Novelty And Significance:** 3
**Empirical Novelty And Significance:** 3
**Recommendation:** 6

**Clarity, Quality, Novelty And Reproducibility:**

I'm not an expert in this subfield, but the core method appears to be novel despite its simplicity (though it could be stated slightly more simply as using randomized sample weights in the loss function rather than per-example gradient weights). It feels likely that prior work has explored use of randomized sample weights, but I don't know of such papers.

The experiments overall are somewhat limited: they're all either at small scale or lacking full hyperparameter tuning (but doing otherwise might require more compute than the authors have). This doesn't make it a bad paper—in fact, it's likely easier to reproduce, and the authors point out that the ImageNet comparison to a very well tuned baseline makes things harder for GradientMix—but it does mean that I would need to replicate the work at a much larger scale before having a good estimate of how useful GradientMix is in practical large-batch settings.

**Strength And Weaknesses:**

Strengths:
- The main goal of the work (better optimization at large batch sizes) is very important to the community.
- The core method is simple and seems to work well, representing meaningful progress on this goal.
- CIFAR-10, CIFAR-100, and ImageNet are all standard image classification benchmarks, and while CIFAR datasets and models are small by contemporary standards they're still good testbeds for rapid iteration.
- The paper is fairly convincing: I think this method is worth trying in my own work.
- The paper relates and compares GradientMix to similar ideas in data augmentation.

Weaknesses:
- The experimental settings for Transformers are quite limited. Multi30k contains only 30k examples, so the entire dataset is around the same size as a single batch in contemporary large-batch Transformer training.
- More generally, contemporary use cases driving the desire for large-batch training are frequently in the regime where each example is visited once (LLMs, recommenders), but none of the experiments cover this case.
- Experimental results are reported only for test accuracy (I think?) but showing both train and test curves would be valuable for understanding the differential impact of GradientMix on optimization and generalization.
- The ImageNet experiment uses a single batch size, but it would be ideal to demonstrate that the critical batch size (the point at which linear convergence scaling wrt batch size ends) is higher for GradientMix than for the baseline; this would require experimental results at multiple batch sizes.
- The paper observes that the advantage of GradientMix is low or reversed at small batch sizes, and makes a suggestion as to the cause without really digging into the question.
- This and other aspects of the picture here could be made clearer by use of the gradient noise scale concept/quantity introduced in McCandlish 2018.

**Summary Of The Paper:**

The paper proposes GradientMix, a regularization method where minibatch gradients are computed as a linear combination of per-example gradients with randomized rather than uniform example weights. The primary application it explores is improving generalization at large batch sizes.

The paper demonstrates GradientMix in a few standard deep learning settings (ConvNets on CIFAR and ImageNet and a sequence-to-sequence Transformer) and shows better generalization at large batch sizes. The authors also provide a theoretical explanation based on GradientMix reducing the trace of the generalized Gauss-Newton matrix, a common approximation of the sharpness of the minimum.

**Summary Of The Review:**

I really appreciate straightforward papers presenting simple but novel improvements to deep learning methods, and that's exactly what I see here. But I would also like to see both more theory (especially discussion of gradient noise scales, as in McCandlish 2018) and more experimental results (at least the train and test losses to show generalization gap).

---

### Official Review · Reviewer_yVcz · 2022-10-31

**Confidence:** 4
**Correctness:** 3
**Technical Novelty And Significance:** 3
**Empirical Novelty And Significance:** 1
**Recommendation:** 5

**Clarity, Quality, Novelty And Reproducibility:**

The paper is well written and easy to follow. The idea is simple and interesting. The authors provide justification that random weights help reduce the trace of the GGN matrix, which is convincing to me.

No code is provided for reproducibility.

**Strength And Weaknesses:**

Strength:
1. The paper introduces an interesting method for improving the accuracy of large batch training. The intuition of reducing trace of GGN matrix makes sense to me and flat minima generalizing better is backed up by several existing works.

Weakness:
1. The empirical improvement looks marginal. For the majority of the experiments, the improvement over the baselines are within 0.4%. It is possible that the gap could be closed with some hyper-parameter tuning.
2. Large batch training for ResNet model has been well studied. (Sun, P., et al., 2019) shows that we can train ResNet-50 on ImageNet in 7.3 mins with a large batch size of 64K, which is already larger than the largest batch size considered in the paper. I think it is more interesting if GradientMix could improve the large batch training of multi-billion parameter GPT model, which is known to suffer from accuracy degradation of large batch and hasn't been studied in the literature.

Sun, P., Feng, W., Han, R., Yan, S., & Wen, Y. (2019). Optimizing network performance for distributed dnn training on gpu clusters: Imagenet/alexnet training in 1.5 minutes. arXiv preprint arXiv:1902.06855

**Summary Of The Paper:**

This paper proposed to incorporate mix-up to the gradient synchronization. Traditionally, the gradients are aggregated by a simple averaging with uniform weight to ensure unbiasedness and low variance. However, this paper argues that random weights following certain distribution pushes the optimizer to minimize the trace of the the generalized Gauss-Newton (GGN) matrix, helping to find a flatter solution than the one obtained via usual average. Since the random weights are reduced to uniform weights in expectation, the convergence of the proposed GradientMix is still guaranteed. The proposed method can be used in conjunction with any optimizers and achieves consistently better performance in training ResNet on CIFAR-10, 100, ImageNet datasets for different batch sizes. The authors also validate the performance on training a small Transformer model on the Multi30k dataset.

**Summary Of The Review:**

This paper introduces a simple and interesting approach for improving the generalization performance. The intuition is backed up by a sketch on how GradientMix tends to minimize the trace of GGN matrix. However, the empirical improvements are limited and slight. Given that ResNet training has been well studies and the proposed achieves very marginal improvement, it is more interesting to see if the proposed method could bring any benefit on training a multi-billion parameter GPT model, which is known be more challenging for large batch training.

---

### Official Review · Reviewer_SMM4 · 2022-11-01

**Confidence:** 3
**Correctness:** 4
**Technical Novelty And Significance:** 2
**Empirical Novelty And Significance:** 2
**Recommendation:** 5

**Clarity, Quality, Novelty And Reproducibility:**

Experiments:
The empirical studies conducted in this paper do not seem to be sufficiently convincing.

- From the experiments conducted I can tell that the improvement by GradientMix is very minor. For examples:
(1) On CIFAR-10 with ResNet-18 the improvement is within 0.1% ; (2) On Imagnet and ResNet-50, the improvement is within 0.2%.
It is hard to tell if it is caused by better hyper-parameter tuning or by the proposed techniques in this paper.

- The coverage of the models and tasks you experimented with is small, which makes it more questionable, e.g., how many tasks the proposed techniques can really generalize to?

- The baseline used in this paper seemed a bit stale IMO. LAMB and LARS are relatively old and weak baselines. I think large-batch training is a rather active field of research. Did you consider comparing it to newer baseline like AdaScale [2]? Also please refer to [2] and see the coverage of experiment i.e., model/task setup.

- How large is the transformer model you used for the language task?


Writing clarity:

Overall I feel this paper is easy to follow. However, the introduction of this paper appears as a combination of a “related work” section and some statements of contributions, which repeats the majority of contents of section 4.
I’d suggest rewriting the introduction (sec. 1) to focus on high-level intuition, e.g., pointing to us what would be a high-level idea of this paper, i.e., at a high level, why would apply gradient mix in large batch training work?

[1] mixup: Beyond Empirical Risk Minimization
[2] AdaScale SGD: A User-Friendly Algorithm for Distributed Training


**Strength And Weaknesses:**

Strength:
- The technical part of the writing is clear and easy to follow
- The idea of using mixed-up gradients to reduce GCN seems sound to me

Weakness:
- The introduction is not well-written. It seems to just repeat the related work section
- The effectiveness of the proposed methods is questionable. From the experiments presented in the paper I can tell that the improvement of the proposed technique provides very minor advantages; and not sure how it can actually help distributed training in practice.

See detailed comments in the next section

**Summary Of The Paper:**

This paper applies the mix-up approach proposed in [1] to the parallel gradients in the data-parallel large-batch training regime and shows that it can provide a minor improvement on the results on validation dataset son two tasks (image classification and machine translation) and three small NN architectures (ResNet-15, ResNet-50, and a transformer model).

**Summary Of The Review:**

The paper applied the mixed-up technique to the distributed gradients in data-parallel large-batch training. According to the experiments presented in the paper, I feel the improvement is rather minor, and am not sure how it can be applied in real large-batching training settings.

---

### Decision · Program_Chairs · 2023-01-20

**Decision:**

Reject

**Justification For Why Not Higher Score:**

Reviewers are borderline negative [5,5,6,5], but the weaknesses mentioned in the summary above are too severe to allow acceptance.

**Justification For Why Not Lower Score:**

N/A

**Metareview: Summary, Strengths And Weaknesses:**

This paper suggests a simple method for improving large-batch training by mixing local gradient. The main advantage of this method is that is very simple to implement. However, there are several weaknesses:
1) Overall improvements seem rather small (e.g., 0.1%-0.2% in ResNet50 ImageNet).
2) The baselines are not up-to-date. One reviewer mentioned Adascale*, but I think there are other baselines missing, e.g. methods from [A] and [B].
3) The experiments seem too limited compared to previous papers (in terms of models/datasets/batch sizes).



*The authors reported Adascale CIFAR results in their response, but that is not sufficient evidence.

[A] Yong Liu, Xiangning Chen, Minhao Cheng, Cho-Jui Hsieh, Yang You, Concurrent Adversarial Learning for Large-Batch Training, ICLR 2022

[B] Jonas Geiping, Micah Goldblum, Phillip E. Pope, Michael Moeller, Tom Goldstein, Stochastic Training is Not Necessary for Generalization, ICLR 2022